# Wild-Type IDH Enzymes as Actionable Targets for Cancer Therapy

**DOI:** 10.3390/cancers11040563

**Published:** 2019-04-19

**Authors:** Elisa Bergaggio, Roberto Piva

**Affiliations:** Department of Molecular Biotechnology and Health Sciences, University of Torino, via Nizza 52, 10126 Torino, Italy; elisa.bergaggio@gmail.com

**Keywords:** isocitrate dehydrogenase (IDH), wild-type IDH inhibitors, combination therapy, non-oncogenic addition, α-ketoglutarate (αKG), reactive oxygen species (ROS), DNA damage

## Abstract

Isocitrate dehydrogenases (IDHs) are enzymes that catalyze the oxidative decarboxylation of isocitrate, producing α-ketoglutarate (αKG) and CO_2_. The discovery of IDH1 and IDH2 mutations in several malignancies has brought to the approval of drugs targeting IDH1/2 mutants in cancers. Here, we summarized findings addressing the impact of IDH mutants in rare pathologies and focused on the relevance of non-mutated IDH enzymes in tumors. Several pieces of evidence suggest that the enzymatic inhibition of IDHs may have therapeutic potentials also in wild-type IDH cancers. Moreover, IDHs inhibition could enhance the efficacy of canonical cancer therapies, such as chemotherapy, target therapy, and radiotherapy. However, further studies are required to elucidate whether IDH proteins are diagnostic/prognostic markers, instrumental for tumor initiation and maintenance, and could be exploited as targets for anticancer therapy. The development of wild-type IDH inhibitors is expected to improve our understanding of a potential non-oncogenic addition to IDH1/2 activities and to fully address their applicability in combination with other therapies.

## 1. Isocitrate Dehydrogenase Enzymes

Isocitrate dehydrogenases (IDHs) are enzymes that catalyze the oxidative decarboxylation of isocitrate, producing α-ketoglutarate (αKG) and CO_2_. In humans, IDHs exist in three isoforms: IDH1, IDH2, and IDH3. IDH1 and IDH2 use nicotinamide adenine dinucleotide phosphate (NADP^+^) as a co-factor and function as homodimers. The genes encoding for IDH1 and IDH2 lie on chromosome 2q34 and on chromosome 15q26.1, respectively. The two enzymes have different cellular localization: IDH1 is situated in the cytosol and the peroxisomes, whereas IDH2 is in the mitochondria [1]. IDH3 is also located inside the mitochondria, although it uses nicotinamide adenine dinucleotide (NAD^+^) as a co-factor. IDH3 forms heterodimers composed by αβ and αγ subunits, that are assembled into the α_2_βγ heterotetramer, which can be further dimerized into a heterooctamer [2]. The IDH3α-subunit is encoded by the *IDH3A* gene (chromosome 15q25.1), the β-subunit by the *IDH3B* gene (chromosome 20p13), and the γ-subunit by the *IDH3G* gene (chromosome Xq28). The reaction catalyzed by IDH3 generates αKG and NADH within the tricarboxylic acid (TCA) cycle and is irreversible. αKG is further metabolized to succinate, while NADH is used by the electron transport chain to generate ATP. Even though IDH1/2 enzymes catalyze the equivalent isocitrate-to-αKG conversion, their reactions are coupled to NADP^+^ reduction and are reversible. The oxidative decarboxylation that converts αKG to isocitrate occurs predominantly in hypoxic conditions producing citrate and acetyl-CoA from glutamine and glutamate. This activity is critical to preserving lipids and cholesterol biosynthesis in hypoxic cells [3,4,5,6,7]. Beyond their role in intermediary metabolism and energy production, IDH enzymes are involved also in redox status regulation. Indeed, NAD(P)^+^/NAD(P)H cofactors are essential for electron transfer in a plethora of cellular functions [8,9,10,11]. Specifically, NADPH secures an adequate pool of reduced glutathione (GSH) [12,13], thioredoxin [14], and catalase tetramers [15], required to counteract the formation of reactive oxygen species (ROS). In addition, αKG enables the activity of αKG-dependent dioxygenases, such as the ten-eleven translocation (TET) family of 5-methylcytosine hydroxylases, the Jumonji-domain containing histone-lysine demethylases (Jmj-KDMs), the AlkB family of dioxygenases, the hypoxia-inducible factor (HIF) prolyl 4-hydroxylases and asparaginyl hydroxylase, and the collagen prolyl and lysine hydroxylases, required for DNA and histone demethylation, DNA repair, HIF degradation, and collagen maturation and folding, respectively [16,17,18,19].

## 2. *IDH*s Genetic Alterations in Cancer

In recent years, this class of enzymes has received much attention, as mutations of *IDH1* and *IDH2* genes have been found in several malignancies, in particular in ~80% of grade II and III astrocytomas, oligodendrogliomas, and oligoastrocytomas and in secondary glioblastomas [20,21,22,23], ~60% of chondrosarcomas [24,25], ~40% of angioimmunoblastic T cell lymphoma [26], ~20% of intrahepatic cholangiocarcinomas [27], ~10% of acute myeloid leukemias [28,29], ~10% of melanomas [30], ~5% of myelodysplastic syndromes and myeloproliferative neoplasms [29,31,32,33], and less frequently in other types of cancers [34,35,36,37].

*IDH1/2* mutations are heterozygous and result in amino acid changes that occur primarily at residue R132 in IDH1 and R140 or R172 in IDH2. The mutant proteins display a new enzymatic activity able to catalyze the NADPH-dependent reduction of αKG to D-2-hydroxyglutarate (D-2HG) [31,38,39]. The consequence is a decrease in αKG and NADPH, associated with the production of the oncometabolite D-2HG and NADP^+^. This has a critical impact on the epigenetic cell status, blocking cellular differentiation by competitively inhibiting αKG-dependent dioxygenases involved in histone and DNA demethylation [28,40,41], together with additional alterations in cellular metabolism, redox state, and DNA repair. The relevance of these mutations and their role in carcinogenesis has been extensively reviewed elsewhere [19,42,43,44,45].

The appreciation of the role of IDH1/2 mutations in oncogenesis and their early occurrence prompted the development of IDH1/2-mutant inhibitors. Recently, the US Food and Drug Administration approved the use of enasidenib (AG-221) and ivosidenib (AG-120) for the treatment of refractory or relapsed acute myeloid leukemia mutated in IDH2 or IDH1, respectively [46,47].

## 3. *IDH*s Genetic Alterations in Genetic Diseases

Mutations in *IDH* genes have been found also in noncancerous diseases. Indeed, *IDH1* or *IDH2* heterozygous mutations have been described in Ollier disease (81% carried *IDH1/2* mutations in their tumors) and Maffucci syndrome (77%), that are usually non-hereditary skeletal disorders [24,48,49]. The Ollier disease is characterized by multiple enchondromas, a benign growth of cartilage within the bones, that may result in bone deformities and fractures. In Maffucci syndrome, multiple enchondromas are combined with red or purplish growths in the skin consisting of tangled blood vessels (spindle cell hemangiomas) [50]. In these disorders, IDH1/2 mutations represent early post-zygotic occurrences, thus generating mosaicism. Compatible with this model, IDH1/2 mutations have been found in cells of enchondromas and hemangiomas, as well as in the bone marrow or blood of a few affected individuals [48]. As described for cancer patients, mutant enzymes produce D-2HG [48]. It has been shown that IDH1/2 mutations contribute to the formation of cartilaginous tumors through the dysregulation of the chondrogenic and osteogenic differentiation of mesenchymal stem cells via gene-specific histone modulation [51]. Considering these observations, mutant IDH-targeted therapy can be suggested as a potential approach to treat these tumors, for which no current effective therapies are available. Future experiments are mandatory to assess the efficacy of IDH inhibitors in these pathologies.

R140 IDH2 mutations have also been found in D-2-hydroxyglutaric aciduria (D-2-HGA) patients [52]. D-2-HGA is a rare neurometabolic disorder, with a wide range of symptoms. Children can be asymptomatic or have developmental delay, epilepsy, hypotonia, cardiomyopathy, brain white matter abnormalities, and dysmorphic features [53]. All affected individuals have consistently increased D-2HG levels in urine, plasma, and cerebrospinal fluid [54]. Approximately 50% of cases display an inherited autosomal recessive pathology, characterized by the homozygous inactivation of the gene *D-2-hydroxyglutarate dehydrogenase* (*D2HGDH)*, and defined as D-2-HGA type I. *D2HGDH* encodes for the D-2-hydroxyglutarate dehydrogenase, which hydrolyzes D-2HG to αKG [55]. The second most common form of D-2-HGA (type II) is considered an autosomal dominant disorder characterized by *IDH2* heterozygous mutations occurring in people with no history of the condition in their family. *IDH2* mosaic mutations have also been observed in a D-2-HGA patient and in an unaffected mother, who was a mosaic carrier [56]. Akbay et al. generated an inducible IDH2 R140Q mouse model that recapitulates the abnormalities observed in D-2-HGA patients. Upon doxycycline withdrawal and extinction of transgene expression, they observed a reduction in serum D-2HG levels accompanied by improved heart function and increased overall survival (OS), indicating that inhibitors of mutant IDH2 may be beneficial in the treatment of D-2-HGA [57].

*IDH1* somatic mosaic mutations were reported also in isolated cases of spondyloenchondromatosis with D-2-hydroxyglutaric aciduria (also known as metaphyseal enchondrodysplasia with 2-hydroxyglutaric aciduria or metaphyseal chondromatosis with D-2-hydroxyglutaric aciduria) [56,58]. This pathology is a very rare skeletal dysplasia with multiple enchondromata in the metaphyses of the long bones associated with dysplastic vertebral bodies. Since *IDH1* mutations were identified in a limited number of patients, further studies are required to dissect the role of IDH1 mutations in this pathology. Furthermore, homozygous loss-of-function mutations in *IDH3A* and *IDH3B* genes have previously been implicated in families exhibiting retinitis pigmentosa, a hereditary neurodegeneration of rod and cone photoreceptors [59,60].

## 4. Downregulation of Wild-Type IDH1 in Cancers

While mutated *IDH1/2* genes have been thoroughly described in cancers, the significance of aberrant expression of these enzymes and their possible therapeutic implications have been partially investigated. Here, we summarize studies reporting on aberrant IDH1, IDH2, and IDH3 expressions in cancers.

For what concerns IDH1, its expression has been found downregulated during early skin tumorigenesis [61]. Specifically, IDH1 expression decreases under exposition to the tumor promoters 12-O-tetra-decanoylphorbol-13-acetate (TPA) and ultraviolet C (UVC) irradiation. Indeed, the authors demonstrated that IDH1 knockdown enhanced TPA efficacy to induce transformation of the skin epidermal promotable JB6 cells. Conversely, IDH1 overexpression represses the pro-oncogenic effect of TPA. The authors speculate that the oxidative stress generated by tumor promoters might contribute to IDH1 inactivation. Indeed, manganese superoxide dismutase overexpression, a mitochondrial antioxidant enzyme, blocks IDH1 decrease. According to this study, the induction of IDH1 activity may serve as a novel chemopreventive strategy.

## 5. Overexpression of Wild-Type IDH1 in Cancers

On the other hand, IDH1 was found overexpressed in numerous cancers. Importantly, several studies indicate that IDH1 overexpression correlates with poor OS in the non-small cell lung carcinoma (NSCLC) patients adenocarcinoma and squamous cell carcinomas [62,63,64,65]. It has been demonstrated that shRNAs targeting IDH1 decreased in vitro and in vivo growth of NSCLC cell lines [62]. Moreover, IDH1 levels in plasma of lung squamous cell carcinoma and lung adenocarcinoma patients were significantly elevated compared to benign lung disease patients and healthy individuals, suggesting IDH1 as a potential plasma biomarker for the diagnosis of NSCLCs [66].

Furthermore, IDH1 is overexpressed in approximately 65% of primary glioblastomas (GBM), in the absence of *IDH1* gene copy number gains or epigenetic activation [65]. IDH1 inactivation with shRNAs or GSK864 (an IDH1 inhibitor) decreases GBM cell growth, promotes a more differentiated tumor cell state, increases apoptosis in response to receptor tyrosine kinase inhibitors, and prolongs the survival of patient-derived xenografts mice. IDH1 inactivation results in decreased αKG and NADPH, with consequent exhaustion of GSH and increased levels of ROS, reduction of lipid biosynthesis, and enhanced histone methylation and differentiation marker expression. In contrast, ectopic expression of IDH1 accelerated the in vivo growth of murine neural stem cells. Therefore, IDH1 upregulation in GMB could represent a common metabolic adaptation to support macromolecular synthesis, aggressive growth, and therapy resistance [65,67].

A gene expression profile analysis of the cancer genome atlas (TCGA) dataset identified IDH1 upregulation in several hematological malignancies, including angioimmunoblastic lymphoma, anaplastic large cell lymphoma, peripheral T cell lymphoma, and diffuse large B cell lymphoma (DLBCL) [65]. Concordantly, IDH1 silencing in a DLBCL cell line decreased αKG and GSH production, with subsequent ROS increase and tumor growth reduction. These effects were associated with enhanced apoptotic susceptibility to the Bruton’s tyrosine kinase (BTK) inhibitor ibrutinib. An independent study demonstrated that high IDH1 expression was correlated to poor prognosis in cytogenetically normal acute myeloid leukemia patients [68]. In addition, increased IDH1 mRNA levels were observed in primary and metastatic pancreatic ductal adenocarcinoma (PDAC) [69]. The authors discovered that nutrient withdrawal initiates an adaptive pro-survival and antioxidant program that renders PDAC cells resistant to additional oxidative insults, such as chemotherapy. This mechanism implicates the activation of the RNA binding protein Hu antigen R/ELAV like RNA binding protein 1 (HuR), which positively regulates IDH1, increasing its antioxidant defense to preserve survival under stress conditions. Indeed, IDH1 overexpression in HuR-deficient PDAC cells was sufficient to fully restore chemoresistance under low nutrient conditions, thus highlighting a potential metabolic vulnerability and therapeutic opportunity.

## 6. Downregulation of Wild-Type IDH2 in Cancers

Numerous reports recognized aberrant IDH2 expression in cancers. Specifically, IDH2 was significantly downregulated in melanomas as compared with nevi. IDH2 reduction was associated with a decreased activity of TET family enzymes and an overall loss of 5-hydroxymethylcytosine (5-hmC), in agreement with the notion that TET proteins are 5-mC DNA hydroxylases requiring αKG as a cofactor. Accordingly, *IDH2* or *TET2* reintroduction in melanoma cells restores the 5-hmC landscape, suppresses melanoma growth, and increases tumor-free survival (TFS) in a zebrafish melanoma model [70].

IDH2 levels were decreased in kidney cancer [64], hepatocellular carcinoma (HCC) [71,72], and gastric cancer (GC) [64,73], compared to normal tissues. Moreover, IDH2 expression was particularly decreased in metastatic HCC and GC compared to those without metastases. The authors suggested that IDH2 down-regulation could promote cell invasion via an NF-ĸB-dependent increase of matrix metalloproteases [72,73]. In addition, low IDH2 expression was associated with worse OS and higher cumulative recurrence rates in HCC. Therefore, IDH2 has been suggested as an independent prognostic marker for OS and time to recurrence in HCC [71].

Expression of IDH2 mRNA and protein was found downregulated also in malignant gliomas, as compared with peripheral non-tumorous brain tissues. The authors demonstrated that miR-183 targets IDH2, decreasing its expression at mRNA and protein levels. Consequently, miR-183 activity results in HIF-1α upregulation, that might contribute to tumorigenesis through enhanced angiogenesis, metabolism, and survival [74,75].

It is well established that IDH2 activity can be regulated by Sirt3, a sirtuin able to deacetylate IDH2, thus increasing its activity [76]. Interestingly, lower Sirt3 protein expression was associated with worse OS in mantle cell lymphoma (MCL) patients. Furthermore, Sirt3 protein expression was reduced in chronic lymphocytic leukemia (CLL) primary samples and in malignant B-cell lines. Lower Sirt3 expression correlates with IDH2 and SOD2 protein hyperacetylation, decreased enzymatic activities, and higher ROS levels. Loss of Sirt3 increases proliferation via ROS-dependent mechanisms, as demonstrated by the rescue obtained with a ROS scavenger [77]. On the contrary, Sirt3 overexpression increases IDH2 and SOD2 activities, decreases ROS levels, glucose consumption, and lactate production, inhibiting cancer cells proliferation. Concordantly, IDH2 knockdown increases B cell proliferation, while its overexpression decreases proliferation.

## 7. Overexpression of Wild-Type IDH2 in Cancers

IDH2 protein expression levels are significantly upregulated in esophageal squamous cell cancer (ESCC) tissues than in paracancerous tissues. Kaplan-Meier analysis showed that IDH2 overexpression in ESCC patients was significantly related to worse OS and progression-free survival (PFS), suggesting IDH2 expression as an independent prognostic marker in these patients. Moreover, IDH2 targeting by shRNAs inhibits proliferation and invasion of human ESCC cell lines, while IDH2 upregulation showed the opposite effects [78]. Furthermore, IDH2 expression was found to be higher in lung cancers as compared to normal lung tissues. High IDH2 levels correlate with poor OS and shorter PFS. IDH2 overexpression in lung cancer cell lines decreased αKG and ROS, and induced HIF1α and Warburg effect, resulting in cell growth increase. IDH2 inhibition reversed these effects, inhibiting cells proliferation [64]. In contrast, it has been shown that IDH2 silencing exacerbates cellular apoptosis induced by the environmental pollutant acrolein in Lewis lung carcinoma cells. The combined cytotoxic effect is explained as a consequence of increased ROS generation and subsequent induction of oxidative damages. Moreover, IDH2 silencing intensified mitochondrial dysfunctions caused by acrolein, with disruption of mitochondrial membrane potential and decreased ATP levels. Accordingly, IDH2 knockout (KO) in mice promoted acrolein-induced lung injury and toxicity, through the disruption of mitochondrial redox status. Indeed, the ROS scavenger N-acetylcysteine shows protective effects against acrolein toxicity in vitro and in vivo [79]. This study highlighted the important role of IDH2 also in preventing cancer formation.

It has been reported that IDH2 levels are up-regulated also in ovarian cancers [64,80]. Moreover, IDH2 overexpression in endometrioid carcinomas of the endometrium has been suggested as a possible marker for differential diagnosis from endometrioid carcinomas of the ovary [81].

Furthermore, publicly available gene expression databases identified high levels of IDH2 expression in prostate, testis, eye, nervous system, and breast cancers [64,82]. Correlation studies revealed that breast and pancreatic cancer patients with high IDH2 levels exhibit worse OS and PFS [64]. However, results in breast cancers are contradictory. Indeed, IDH2 acetylation (that reduces IDH2 dimer formation and thus its enzymatic activity) was elevated in high-risk luminal B relative to low-risk luminal A patients [83]. Moreover, expression of an acetylation mimetic IDH2 mutant (IDH2K413Q) in the breast cancer cell line MCF7 was associated with increased cellular ROS and glycolysis, decreased mitochondrial respiration capacity and adenosine triphosphate (ATP) production, promotion of cell transformation and tumorigenesis in nude mice [83]. Previously, it was suggested that Sirt3 silencing, consequent IDH2 hyper-acetylation, and inactivation, slightly reduced viability and increased the cytotoxicity of cisplatin and tamoxifen in breast cancer cell lines, due to an increase in ROS production. Concordantly, a higher Sirt3 expression is related to a poorer prognosis in grade 3 estrogen receptor-positive breast cancer [84].

IDH2 gene expression was found significantly downregulated in early stage (in situ carcinoma) but upregulated in advanced stage (infiltrating carcinoma) colorectal cancer (CRC) compared to peritumor tissue. Accordingly, IDH2 silencing in a CRC cell line significantly inhibited cell growth [85]. Even though, IDH1/2 expression levels have not been correlated to prognosis, patients with ‘imbalanced’ IDH1 and IDH2 expression (i.e., IDH1^high^; IDH2^low^, or IDH1^low^; IDH2^high^) had a shorter disease-free survival and OS compared to patients with a ‘balanced’ IDH expression (i.e., IDH1^high^; IDH2^high^, or IDH1^low^; IDH2^low^) [86]. However, further studies are required to corroborate the hypothesis by which imbalanced IDH1/IDH2 expression results in D-2HG overproduction and consequent oncogenic effects.

Finally, it has been suggested that IDH2 expression in the tumor microenvironment can influence tumorigenesis. Indeed, it was demonstrated that tumorigenesis of B16F10 melanoma cells was strongly reduced when cells are implanted in IDH2-deficient (IDH2^−/−^) mice. This was associated with a significant increase of ROS and oxidative stress alongside with down-regulation of angiogenesis markers in the tumor and the stroma. Tumor growth was further diminished upon exposure to ionizing radiation. These results indicate that IDH2-mediated microenvironment changes in redox status may contribute to cancer progression [87].

## 8. Wild-Type IDH3 Downregulation/Overexpression in Cancer

It was observed that high IDH3α expression is associated with poor postoperative OS in lung and breast cancer patients. IDH3α overexpression in cervical and lung adenocarcinoma cell lines reduces αKG levels, with consequent HIF-1α increased stability and transactivation activity [88]. Consistent with this, the cell-permeable form of αKG abrogates IDH3-mediated activation of HIF-1α. As expected, IDH3α overexpression in HeLa cells increases tumor grow in vivo. In contrast, HIF-1 inhibition significantly suppresses IDH3α-mediated promotion of tumor growth and IDH3α silencing delays tumorigenesis by suppressing HIF-1-mediated Warburg effect and angiogenesis. Interestingly, IDH3α expression and HIF-1 activity are upregulated as a result of cellular immortalization and transformation by the human papillomavirus 18 *E6* and *E7* genes and *K-Ras G12V* gene in mouse embryonic fibroblasts. However, IDH3β silencing significantly suppresses the positive impact of IDH3α overexpression on HIF-1 activity, which indicated that IDH3α functions, at least in part, as a component of the IDH3 heterotetramer to activate HIF-1 [88].

Moreover, IDH3α knockdown decreases ATP levels and cell growth and enhance neuronal differentiation of an embryonic carcinoma cell line [89].

## 9. IDH Inhibition in Cancer Enhances Responsiveness to Canonical Therapies

The above-described data suggest that IDH1/2/3 can be used as biomarkers for diagnosis, prognosis prediction, and target of therapy in several neoplastic diseases. Additional studies indicate that IDH1/2 inhibition could increase the efficacy of conventional cancer therapies. These observations are in line with the understanding that cancer cells depend on the activities of genes and pathways which are not required to the same degree for the viability of normal cells, a concept known as non-oncogene addiction [90]. Nowadays, targeting IDH enzymes for a therapeutic purpose is extremely intriguing given the fact that enhancement of responsiveness to canonical therapies could occur in conditions of deregulated IDH expression, as well as when IDH is expressed at normal levels.

### 9.1. Chemotherapy

It has been observed that IDH1 inhibition sensitizes defined types of cancers to chemotherapy. For instance, IDH1 knockdown in GBM cells increases the efficacy of the alkylating agent/glutathione reductase inhibitor bis-chloroethylnitrosourea (BCNU), which is clinically used as biodegradable wafers after surgical resection, and to the further addition of aminooxyacetate, a transaminase and glutaminolysis inhibitor. Indeed, treatment with BCNU decreases GSH levels which are further depleted in IDH1 silenced cells. The fact that the reducing effect is causative of the higher response is proved by the rescue obtained after the addition of a ROS scavenger. The same cells treated with the alkylating agent temozolomide doesn’t enhance its activity, probably because it is not an equally potent inducer of oxidative stress in comparison to BCNU [91]. Although, several groups report that cells overexpressing IDH1 or IDH2 are more resistant to high doses of temozolomide [91,92,93].

As previously described, response to chemotherapy in nutrient-deprived PDAC cells is modulated by the RNA binding protein HuR, which positively regulates IDH1 increasing ROS clearance. Notably, it has been demonstrated that *HuR* gene deletion abrogates the subcutaneous engraftment of PDAC cell in nude mice. Conversely, exogenous expression of IDH1 completely rescues the growth of *HuR*^−/−^ PDAC cells in vivo. Moreover, IDH1 knockdown further increases ROS levels induced by nutrient withdrawal (either glucose or glutamine) or chemotherapy treatment. Consequently, IDH1 depleted cells are more sensitive to gemcitabine treatment, particularly under glucose deprivation, showing an increase in DNA damage. Accordingly, IDH1 overexpression reduces ROS levels and protects against DNA damage induced by gemcitabine [69].

### 9.2. Radiotherapy

It is known that several biomolecules, such as GSH and thioredoxin, require NADPH to mitigate the oxidative stress induced by ionizing radiation [94]. Moreover, NADPH is necessary for the synthesis of deoxynucleotides, essential to repair radiation-induced DNA damages.

The important role of IDH1 in defense against radiation-induced oxidative injury has been highlighted by several studies. Experiments performed in NIH3T3 cells showed that exposure to UV radiation increases IDH1 activity. Importantly, IDH1 silencing sensitizes to UVB radiation-cell killing with an increase in lipid peroxidation, protein oxidation, oxidative DNA damage, and intracellular peroxide generation. Conversely, IDH1 overexpression enhanced resistance against UV radiation [95].

Although IDH1 overexpression appears not to protect GBM cells to radiotherapy [92], a recent study proved that IDH1 inhibition radiosensitizes GBM cells. The authors demonstrated that radiation further increases the already high expression of IDH1 in GBM and that IDH1 silencing significantly reduced NADPH, deoxynucleotides, and GSH levels, improving radiotherapy response in a murine xenograft model of human GBM. Conversely, IDH1-mediated radiosensitization can be reversed by deoxynucleotide precursors or antioxidants [96]. This mechanism is not exclusive for IDH1 but has been described also for IDH2. As such, IDH2 silencing decreased NADPH and GSH levels resulting in enhanced ROS production and ionizing radiation-induced autophagy in glioma cells [97].

### 9.3. Photodynamic Therapy

Photodynamic therapy is a treatment that uses a light-activated molecule (photosensitizer). The exposure of this molecule to a specific wavelength of light results in ROS production, able to directly induce cellular damage and kill nearby cells [98]. This technique is approved for the treatment of several cancers. A study described that IDH1 silencing enhances the toxicity of the sensitizer rose bengal in the acute myeloid leukemia cell line HL-60, as a consequence of decreased NADPH and GSH levels, and increased ROS production. In contrast, cells overexpressing IDH1 are more resistant to the apoptotic effect mediated by the photosensitizer [99].

### 9.4. Small Molecule Inhibitors

Last decades have brought to the fore targeted cancer therapies and personalized medicine, with a considerable improvement in tumor management [100]. To further enhance the potency of these drugs, several combinatorial treatments are under investigation. A considerable number of studies demonstrated that the efficacy of specific targeted therapies is increased by the combination with IDH inhibitors. For instance, IDH1 silencing specifically increases erlotinib (an EGFR, epidermal growth factor receptor, inhibitor) efficacy in patient-derived glioma-initiating cells (GICs) carrying *EGFR* amplification [65]. Correspondingly, neural stem cells ectopically expressing IDH1 exhibited reduced apoptosis in response to erlotinib. It is suggested that receptor tyrosine kinases (RTKs) inhibitors increase IDH1 production in glioma cells via an RTK-PI3K-Akt-FoxO6 signaling axis. IDH1 induction, leading to lipid biosynthesis and decreased ROS production, has been explained as an adaptive response of glioma cells to growth factor receptor inhibition. Indeed, treatment of erlotinib-primed cells with cell-permeable αKG or the fatty acid palmitate plus the cholesterol precursor mevalonate protected cells from the pro-apoptotic effects of IDH1 silencing, suggesting that reduced fatty acid and cholesterol biosynthesis contribute to the effect mediated by IDH1 knockdown. Consistently, IDH1 silencing affects NADPH and GSH synthesis increasing ROS levels. Treatment of IDH1 silenced cells with ROS scavengers, αKG, or overexpression of cytoplasmic NADPH-generating malic enzyme 1, reduced caspase activation in response to erlotinib treatment, confirming the fundamental role of the NADPH/NADP^+^ ratio for cell survival [65]. Importantly, the same mechanism has been shown to occur also with other tyrosine kinase (TK) inhibitor cocktails. Indeed, IDH1 silencing in glioma cells that display co-activation of multiple RTKs (EGFR, the HGFR family member MSPR, and PDGFRs) enhances the efficacy of inhibitor combinations involving erlotinib, imatinib, and SU11274 [65]. Moreover, IDH1 knockdown increased the response to ibrutinib in a subcutaneous xenograft model of DLBCL, by attenuating αKG and GSH production and increasing ROS levels [65].

As previously described, IDH1 silencing increases the toxicity of BCNU/aminooxyacetate combination in glioma cells. In contrast, IDH1 overexpression renders glioma cells more resistant to this combo. This synergy is explained through the further inhibition of αKG production both by IDH1 knockdown and by the block of glutaminolysis pathway [91].

Recently, it has been demonstrated that IDH2 inhibition enhances proteasome inhibitors (PIs) responsiveness in multiple myeloma (MM), mantle cell lymphoma, and Burkitt lymphoma cell lines, as well as in primary CD138^+^ cells from MM patients [101]. The combination significantly decreases tricarboxylic acid cycle activity and ATP levels, as a consequence of enhanced IDH2 enzymatic inhibition. Specifically, this occurs through the inhibition of the NF-κB/NAMPT/SIRT3/IDH2 pathway. Consistently, the combination of PIs with either NAMPT or SIRT3 inhibitors impaired IDH2 activity and increased MM cell death [101,102]. Finally, inducible IDH2 silencing enhanced the therapeutic efficacy of PI carfilzomib in a subcutaneous xenograft model of MM, resulting in inhibition of tumor progression and extended survival [101].

## 10. Concluding Remarks and Future Perspectives

IDH enzymes are known since decades, however, the discovery of IDH1 and IDH2 mutations in several malignancies has brought to the limelight a new consideration of their physiological and pathological functions. This interest led to a better characterization of IDHs role in tumorigenesis and to the approval of drugs targeting IDH1/2 mutants in cancers. Conversely, less is known about the impact of IDH mutants in rare pathologies or the relevance of non-mutated IDH enzymes in cancers. Here, we have summarized findings addressing these two relevant issues.

Even though several genetic diseases carrying somatic or germline IDH mutations have been described (Table 1), the pathogenetic impact of these mutations has to be fully addressed. Moreover, additional studies are required to understand the efficacy of treatments targeting mutated IDH in these pathologies.

Several pieces of evidence suggest that the enzymatic inhibition of IDH1/2 may have therapeutic potentials also in IDH1/2 wild-type cancers. Indeed, alterations of IDH1/2 expression were described in numerous studies. However, proofs are limited and contradictory results have been described. At present, it is not clear whether IDH1/2/3 activities may have a tumorigenic or tumor suppressive role. Overall, it emerges that the impact of IDH1/2/3 expression levels is strongly dependent on tumor type (Table 2). These data are not unexpected, considering the clinical implication of IDH1/2 mutations. For instance, IDH1/2 mutations are a favorable prognostic index in GBM, while they confer adverse prognosis in cytogenetically normal acute myeloid leukemia [103,104]. However, cited studies frequently lack a complete analysis to confirm the impact of IDHs deregulation in tumors. In principle, IDH proteins expression should be characterized in a large series of tumor samples with documented clinical, pathological, and molecular information. Moreover, the measurement of IDH enzymatic activities would be more appropriate to pinpoint differences between tumor and normal tissues. In addition, further studies are mandatory to elucidate whether, how, and to what extend IDH proteins are diagnostic/prognostic markers, instrumental for tumor initiation and maintenance, and could be exploited as targets for anticancer therapy.

Interestingly, IDH1/2 can be inhibited to enhance responsiveness to canonical cancer therapies, such as chemotherapy, radiotherapy, photodynamic therapy, and small molecule inhibitors. Importantly, this combinatorial effect occurs not only when IDH have an altered expression, but also when they are expressed at normal levels, increasing the applicability of IDH as potential targets. The synergic effect is explained through a decrease in αKG, ATP, lipid, and dNTP synthesis and an increase in ROS, with consequent DNA damage, lipid peroxidation, and protein oxidation (Figure 1). These effects exacerbate the damage caused by canonical therapies increasing cell death, compared to single treatments. To date, inhibitors directed against wild-type IDHs are not present on the market. Preliminary data of IDH1/2 targeting showed a specific cytotoxic effect against tumor cells, thus supporting the applicability of wild-type IDH inhibitors [65,101]. Moreover, nanovectors can be utilized to selectively deliver IDHs inhibitor to tumor cells. Interestingly, it was observed that nanoparticles themselves can affect IDH expression, for instance, SiO_2_ nanoparticles inhibit IDH expression, whereas TiO_2_ nanoparticles have the opposite effect, expanding the possibilities to modulate IDH activity [105,106]. The development of IDH inhibitors will largely improve our understanding of a potential non-oncogenic addition to IDH1/2 activities and to fully address the applicability of wild-type IDH inhibitors in combination or not with other therapies.

In conclusion, all these studies clearly put on stage IDHs as new potential targets. Further pre-clinical experiments could bring to the application of new drugs for IDH1/2 wild-type patients refractory to conventional therapy.

## Figures and Tables

**Figure 1 cancers-11-00563-f001:**
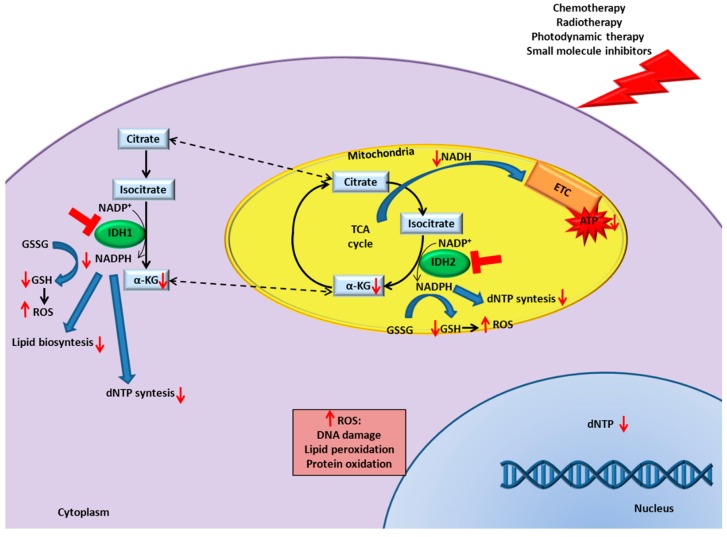
IDH inhibition enhances canonical cancer therapies efficacy. Schematic representation of the underlying mechanisms that increase the response to cancer therapies in presence of IDH1/2 inhibition.

**Table 1 cancers-11-00563-t001:** Isocitrate dehydrogenase (IDH) mutations in genetic diseases.

IDH1 mut.	IDH2 mut.	IDH3 mut.
Ollier disease [24,47,50]	Ollier disease [24,47]	Retinitis pigmentosa [59,60]
Maffucci syndrome [24,47,50]	Maffucci syndrome [47]	
Spondyloenchondromatosis with D-2-hydroxyglutaric aciduria [55,57]	D-2-hydroxyglutaric aciduria [51,55]	

**Table 2 cancers-11-00563-t002:** IDH expression deregulation in cancers. IDH expression levels in the indicated cancers compared to healthy/benign disease tissues, if not otherwise specified. OS = putative oncosuppressive role; OG = putative oncogenic role.

**IDH1 Levels**		
**Downregulated**	**Overexpressed**	**Not Specified**
Early skin tumorigenesis (OS) [61]	Lung squamous cell carcinoma (OG) [62,65,66]	Acute myeloid leukemia (OG) [68]
	Lung adenocarcinoma (OG) [62,63,65,66]	
	Primary glioblastoma (OG) [65,67]	
	Angioimmunoblastic lymphoma [65]	
	Anaplastic large cell lymphoma [65]	
	Peripheral T cell lymphoma [65]	
	Diffuse large B cell lymphoma (OG) [65]	
	Pancreatic ductal adenocarcinoma (OG) [69]	
Imbalance of IDH1/2 in colorectal cancer (OG) [86]		
**IDH2 Levels**		
**Downregulated**	**Overexpressed**	**Not Specified**
Melanoma (OS) [70]	Esophageal squamous cell cancer (OG) [78]	Lewis lung carcinoma (OS) [79]
Kidney cancer [64]	Lung cancer (OG) [64]	Pancreatic cancer (OG) [64]
Hepatocellular carcinoma (OS) [71,72]	Ovarian cancer [64,80]	
Gastric cancer (OS) [64,73]	Endometroid carcinomas of the endometrium vs. endometroid carcinomas of the ovary [81]	
Glioblastoma [75]	Prostate cancer [82]	
Grade III glioma [75]	Testis cancer [82]	
Mantle cell lymphoma (OS) [77]	Eye cancer [82]	
Chronic lymphocytic leukemia [77]	Nervous cancer [82]	
Acute lymphocytic leukemia [77]	Breast cancer (OG/OS) [64,82,83,84]	
Burkitt’s lymphoma (OS) [77]	Infiltrating colorectal cancer (OG) [85]	
High-risk luminal B vs. low-risk luminal A breast cancer [83]		
In situ colorectal cancer [85]		
Imbalance of IDH1/2 in colorectal cancer (OG) [86]		
**IDH3 Levels**		
**Downregulated**	**Overexpressed**	**Not Specified**
		Lung cancer (OG) [88]
		Breast cancer (OG) [88]
		Cervical adenocarcinoma (OG) [88]
		Embryonic carcinoma (OG) [89]

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
