# Peer review of "Wild-Type IDH Enzymes as Actionable Targets for Cancer Therapy"

_cancers, 2019, doi:10.3390/cancers11040563_

Round 1

Reviewer 1 Report

This review summarizes the literature on isocitrate dehydrogenases (IDHs) as potential targets for cancer therapy. These citric acid cycle enzymes catalyze the oxidative decarboxylation of isocitrate to produce α-ketoglutarate (αKG) and CO2. However, in certain forms of cancer, arginine residues in the active sites of these enzymes have been mutated. This causes the enzymes to produce D-2-hydroxyglutarate (2HG) instead of α-ketoglutarate. Ivosidenib and enasidenib are FDA approved drugs that are used to treat acute myeloid leukemia (AML), a disease that possesses the mutated forms of isocitrate dehydrogenase. This review focuses on the cancer cell types and conditions that produce a higher concentration (“overexpression”) or lower concentration (“down regulation”) of wild type IDHs enzymes inside cells. The work suggests that inhibiting the wild type enzymes can enhance standard forms of cancer therapy including, chemotherapy, radiotherapy, photodynamic therapy and small molecule inhibitors.

            Since wild type IDHs are found in both normal and cancer cells, “targeting” these enzymes with and inhibitor would not selectively kill cancer cells. Without a sharper focus and a clearer rational, this review will be of limited value to the scientific community.    

Comments/suggestions

The authors suggest that the effects of radiotherapy and photodynamic therapy will somehow be enhanced if wild type IDHs are inhibited (presumable with a small molecule inhibitor). Since radiotherapy and photodynamic therapy ultimately generate ROS which indiscriminately chemically modify biomolecules in their vicinity, it is not clear how these established therapies will be enhanced if IDHs with bound inhibitors are present. What is the connection between the mechanism of these well established cancer treatment techniques and the presence of inhibited IDHs? The authors discuss the concentrations of wild type IDHs in cells (overexpression/down regulation) but it is not clear how this, i.e. the amount of IDHs that is present in the cell, would influence the effects of radiotherapy and photodynamic therapy at the molecular level.    

This manuscript is devoid of chemistry and any discussion about how an inhibitor would interact with wild type IDHs. A logical starting point would be to give the chemical structures of ivosidenib and enasidenib (and maybe other drug candidates as well) and outline how these actively used drugs are believed to interact with the mutated forms of IDHs, highlighting the role of the guanidinium residue in the active site. Since the drugs have relatively simple structures and a number of crystal structures of the enzymes have been published, this should be possible. What are their thoughts on how to inhibit the wild type IDHs with a small molecule from the structural point of view?  

Some of the sentences and passages in this manuscript are very opaque. Improved English and scientific logic are warranted. For example, in lines 292-295 they say, “Experiments performed in NIH3T3 cells showed that exposure to UV radiation increases IDH1 activity. Importantly, IDH1 KD sensitizes to UVB radiation-cell killing with an increase in lipid peroxidation, protein oxidation, oxidative DNA damage, and intracellular  peroxide generation. Conversely, IDH1 overexpression enhanced resistance against UV radiation [94].” The first sentence says that when you treat cell with radiation, more IDH1 is made (they say, “increase IDH1 activity”). The last sentence says that making more IDH1 makes cells more resistant to radiation. I believe that the middle sentence says that more IDH1 in the cell makes radiation treatment more effective. The take home message of this passage is difficult to identify. There are others like this in the manuscript.

Starting on line 378 they say, “Interestingly, IDH1/2 can be inhibited to enhance responsiveness to canonical cancer therapies, such as chemotherapy, radiotherapy, photodynamic therapy, and small molecule inhibitors.” What is the meaning of, “can be inhibited to enhance responsiveness”? Do the authors want to say that when IDH1/2 are inhibited, other forms of cancer chemotherapy are enhanced?

Author Response

REPLY TO REVIEWER #1

We thank the Reviewer for rising up important issues and for his insightful and constructive comments.

…The work suggests that inhibiting the wild type enzymes can enhance standard forms of cancer therapy including, chemotherapy, radiotherapy, photodynamic therapy and small molecule inhibitors. 

Since wild type IDHs are found in both normal and cancer cells, “targeting” these enzymes with an inhibitor would not selectively kill cancer cells. Without a sharper focus and a clearer rational, this review will be of limited value to the scientific community.  

We agree with Reviewer #1 that inhibiting wild-type IDHs might appear not specific for tumor cells. However, in many instances, the proper functioning of a non-mutated gene has been shown to enhance the survival of cancer cells, a phenomenon described as non-oncogenic addition (1). As a matter of fact, tumor cells often display increased levels of various stresses. Therefore, it has been proposed that inhibition of stress-reducing pathways would increase stress to critical levels. This approach is expected to specifically affect tumor cells while sparing normal ones.

In line with this paradigm, several papers demonstrated that cancer cells are more sensitive to IDHs inhibition (2-4). In addition, both IDH1 and IDH2 null mice are healthy, fertile, with minor or late toxic effects, thus supporting the applicability of wild-type IDH inhibitors (5-8).  

We are aware that the success of targeting non-oncogenic additions depends on the severity of side effects. Therefore, it is of utmost importance to determine the difference in sensitivity towards IDH inhibition between malignant and normal tissues in preclinical in-vivo experiments. Moreover, if IDH inhibition will confirm particularly effective against cancer cells, we can envisage to selectively deliver IDH inhibitors to target cells through nanovectors.

We apologize if this issue was not clearly addressed in the original manuscript. We have now modified the text, accordingly to the Reviewer's suggestion.

The authors suggest that the effects of radiotherapy and photodynamic therapy will somehow be enhanced if wild type IDHs are inhibited (presumable with a small molecule inhibitor). Since radiotherapy and photodynamic therapy ultimately generate ROS which indiscriminately chemically modify biomolecules in their vicinity, it is not clear how these established therapies will be enhanced if IDHs with bound inhibitors are present. What is the connection between the mechanism of these well-established cancer treatment techniques and the presence of inhibited IDHs? The authors discuss the concentrations of wild type IDHs in cells (overexpression/down regulation) but it is not clear how this, i.e. the amount of IDHs that is present in the cell, would influence the effects of radiotherapy and photodynamic therapy at the molecular level.

A major challenge in developing new anti-cancer therapies is to identify compounds with a sufficient therapeutic window causing tumor regression with minimal side effects on normal tissues. The principles of non-oncogene addition and combination therapy protocols propose that inhibition of stress-reducing pathways would increase the level of stress to critical levels. These approaches are designed to specifically affect tumor cells by exploiting their vulnerability. This concept has been explored in several clinical contexts, and the most successful example is the use of poly (ADP-ribose) polymerase inhibitors in tumors with mutations in BRCA. 

In the present manuscript, we critically analyzed studies supporting the synergy between canonical therapies and IDH inhibition in several cancer models. Specifically, we described that IDH inhibition could exacerbate the damage caused by canonical therapies increasing cell death, compared to single treatments. These effects, depicted in Figure 1, are explained through a decrease in αKG, ATP, lipid, and dNTP synthesis and an increase in ROS, with consequent DNA damage, lipid peroxidation, and protein oxidation. Moreover, dNTP synthesis reduction after IDH inhibition decreases DNA repair capability, highly required by cells subjected to chemio/radio damage. Importantly, combinatorial effects occur not only when IDH have an altered expression, but also when they are expressed at normal levels, increasing the applicability of IDH as potential targets. 

This manuscript is devoid of chemistry and any discussion about how an inhibitor would interact with wild type IDHs. A logical starting point would be to give the chemical structures of ivosidenib and enasidenib (and maybe other drug candidates as well) and outline how these actively used drugs are believed to interact with the mutated forms of IDHs, highlighting the role of the guanidinium residue in the active site. Since the drugs have relatively simple structures and a number of crystal structures of the enzymes have been published, this should be possible. What are their thoughts on how to inhibit the wild type IDHs with a small molecule from the structural point of view?

We apologize with the Reviewer #1 if the manuscript does not report chemical structures of IDH inhibitors. We avoided going deep into the chemistry of mutant IDH inhibitors, as this was not the focus of our review. Having spent only a few words on the relevance of the mutated forms of IDHs, dwelling on mutant IDH inhibitors would probably be off topic. Moreover, the majority of IDH mutant inhibitors, such as ivosidenib and enasidenib, are not able to efficiently inhibit wild-type IDH proteins. A comprehensive analysis of IDH inhibitors has been published by Urban DJ et al (9). To the best of our knowledge, IDH inhibitors specific for the wild-type forms have not yet been developed. Our review is aimed to highlight the role of wild-type IDHs in tumors, critically analyze relevant studies, and encourage the development of specific IDH inhibitors that could be promising for specific cancer therapies.

Some of the sentences and passages in this manuscript are very opaque. Improved English and scientific logic are warranted. For example, in lines 292-295 they say, “Experiments performed in NIH3T3 cells showed that exposure to UV radiation increases IDH1 activity. Importantly, IDH1 KD sensitizes to UVB radiation-cell killing with an increase in lipid peroxidation, protein oxidation, oxidative DNA damage, and intracellular peroxide generation. Conversely, IDH1 overexpression enhanced resistance against UV radiation [94].” The first sentence says that when you treat cell with radiation, more IDH1 is made (they say, “increase IDH1 activity”). The last sentence says that making more IDH1 makes cells more resistant to radiation. I believe that the middle sentence says that more IDH1 in the cell makes radiation treatment more effective. The take home message of this passage is difficult to identify. There are others like this in the manuscript.

We apologize with the Reviewer if our manuscript was not sufficiently clear.

To improve the logic understanding of the specified sentences, the acronym “KD” has been replaced by the full form “knockdown”. The correct interpretation is: UV induces IDH1; IDH1 silencing increases UV-mediated cell death; IDH1 overexpression increases resistance to radiation.

Starting on line 378 they say, “Interestingly, IDH1/2 can be inhibited to enhance responsiveness to canonical cancer therapies, such as chemotherapy, radiotherapy, photodynamic therapy, and small molecule inhibitors.” What is the meaning of, “can be inhibited to enhance responsiveness”? Do the authors want to say that when IDH1/2 are inhibited, other forms of cancer chemotherapy are enhanced?

The interpretation is correct. As reported in several papers, IDH1 or IDH2 inhibition can increase the efficacy of current cancer treatments, such as chemotherapy, radiotherapy, photodynamic therapy, and small molecule inhibitors.

References

1.    Luo J et al. Principles of Cancer Therapy: Oncogene and Non-oncogene Addiction. Cell. 2009;136: 823–837.

2.    Calvert  AE et al. Cancer-Associated IDH1 Promotes Growth and Resistance to Targeted Therapies in the Absence of Mutation. Cell Reports. 2017;19:1858-187.

3.    Bergaggio E et al. IDH2 inhibition enhances proteasome inhibitor responsiveness in hematological malignancies. Blood. 2019;133:156–167.

4.    Park Jet al.Idh2Deficiency Exacerbates Acrolein-Induced Lung Injury through Mitochondrial Redox Environment Deterioration. Oxid Med Cell Longev.2017;2017:1–13.

5.    Itsumi M et al. Idh1 protects murine hepatocytes from endotoxin-induced oxidative stress by regulating the intracellular NADP(+)/NADPH ratio. Cell Death Differ. 2015;22:1837–1845.

6.    Jing Ye et al. IDH1 deficiency attenuates gluconeogenesis in mouse liver by impairing amino acid utilization. PNAS. 2017;114:292-297.

7.    Lee SJ et al. Amelioration of late-onset hepatic steatosis in IDH2 deficient mice. Free Radical Research.2017;51:368-374.

8.    Lee SJ, et al. Increased obesity resistance and insulin sensitivity in mice lacking the isocitrate dehydrogenase 2 gene. Free Radic Biol Med.2016;99:179-188.

9.    Urban DJ et al. Assessing inhibitors of mutant isocitrate dehydrogenase using a suite of pre-clinical discovery assays. Sci. Rep.2017;7:12758.

Reviewer 2 Report

I commend the Authors for the exhaustive review of the literature on IDH enzymes gene dysregulation and really appreciate the original angle they provide in their piece.

I would have appreciated a bit more insight into the mechanisms by which up- or down-regulation of IDH genes impact tumor progression, but I am absolutely satisfied by their interpretation.

Very minor details to correct:

Line 184: "It's" was misspelled for Its, altogether

Line 216: you may want to consider "publicly available" instead of "public"

Line 219 (and following): you describe the effect of a acetylation-mimetic mutant for IDH, without introducing the 1) IDH can be acetylated and 2) acetylation reduces activity, which are mentioned however little later in the text. I would consider moving swapping paragraphs.

Line 314: add reference

Paragraph 317-329: lots of data and only one reference at the end. It is not clear if the data are related to the one paper cited. If so, maybe add a citation in between, for clarity.

Line 327: I am confused. Adding aKG may not have an effect on NADPH/NADP+ ratio (I agree this is the case for ROS scavengers and over expression of ME1, but aKG does not necessarily supports your conclusion).

Overall, very good job!!

Author Response

REPLY TO REVIEWER #2

I commend the Authors for the exhaustive review of the literature on IDH enzymes gene dysregulation and really appreciate the original angle they provide in their piece.

I would have appreciated a bit more insight into the mechanisms by which up- or down-regulation of IDH genes impact tumor progression, but I am absolutely satisfied by their interpretation.

Very minor details to correct.

We thank Reviewer #2 for his appreciation of the manuscript.

Line 184: "It's" was misspelled for Its, altogether

The typo has been corrected. 

Line 216: you may want to consider "publicly available" instead of "public" 

We thank for the suggestion, we replaced "public" with "publicly available".

Line 219 (and following): you describe the effect of a acetylation-mimetic mutant for IDH, without introducing the 1) IDH can be acetylated and 2) acetylation reduces activity, which are mentioned however little later in the text. I would consider moving swapping paragraphs.

We thank the Reviewer for his suggestion. 

Regulation of IDH2 by SIRT3 is described in section 6 of the manuscript: “It’s well established that IDH2 activity can be regulated by Sirt3, a sirtuin able to deacetylate IDH2, thus increasing its activity [76].” 

To improve the comprehension of the text, the two sentences have been swapped.

Line 314: add reference 

The relevant reference has been included.

Paragraph 317-329: lots of data and only one reference at the end. It is not clear if the data are related to the one paper cited. If so, maybe add a citation in between, for clarity. 

Data are from a single study. We added the reference also at the beginning and in the middle of the cited results. 

Line 327: I am confused. Adding aKG may not have an effect on NADPH/NADP+ ratio (I agree this is the case for ROS scavengers and over expression of ME1, but aKG does not necessarily supports your conclusion). 

We agree with the Reviewer that the connection is poorly explained. According to the study by Ranji Singh et al, we can speculate that aKG addition fuels TCA cycle with consequent NADH production, which in turn can transfer H+to the NADP+, forming NADPH (Ranji Singh, at al., A Novel Strategy Involved Anti-Oxidative Defense: The Conversion of NADH into NADPH by a Metabolic Network, PLoS ONE, 2008; 3(7): e2682).

Overall, very good job!!

Reviewer 3 Report

- A comprehensive review

IDH and IDH1/2 are sometimes written in Italic format and sometimes in normal format (i.e. line 53, 71 and 81). Consists format is required.  

It is recommended to mention the effect of nanoparticles such as SiO2 nanoparticles on the expression of IDHs

Author Response

REPLY TO REVIEWER #3

A comprehensive review

We thank Reviewer #3 for his appreciation of the manuscript and for his constructive comments.

IDH and IDH1/2 are sometimes written in Italic format and sometimes in normal format (i.e. line 53, 71 and 81). Consists format is required.  

We maintained the italic format to indicate the gene to distinguish it from the protein.

It is recommended to mention the effect of nanoparticles such as SiO2 nanoparticles on the expression of IDHs

We thank the reviewer for the suggestion. We added the role of nanoparticles in the modulation of IDHs in the concluding remarks section of the revised manuscript.

Reviewer 4 Report

This a significant and well written review. There are some minor concerns needed be addressed or revised before it is published.

Line 97, D2HGDH, defined as D-2-HGA à D2HGDH, and defined as D-2-HGA

LINE 268, IDH1 inhibition sensitize defined à IDH1 inhibition sensitizes defined

Line 566, The reference might be cited inconsistently with text. The reference is lung cancer related but the lines 150 and 151 in the main text states differently regarding angioimmunoblastic lymphoma, anaplastic large cell lymphoma, peripheral T cell lymphoma, and diffuse large B cell lymphoma…

 Line 574, Journal name Plos One is missing.

Author Response

REPLY TO REVIEWER #4

This a significant and well written review. There are some minor concerns needed be addressed or revised before it is published.

We thank Reviewer #4 for the recognition of the manuscript.

Line 97, D2HGDH, defined as D-2-HGA à D2HGDH, and defined as D-2-HGA 

The typo has been corrected. 

LINE 268, IDH1 inhibition sensitize defined à IDH1 inhibition sensitizes defined

The typo has been corrected. 

Line 566, The reference might be cited inconsistently with text. The reference is lung cancer related but the lines 150 and 151 in the main text states differently regarding angioimmunoblastic lymphoma, anaplastic large cell lymphoma, peripheral T cell lymphoma, and diffuse large B cell lymphoma…

The mentioned study (Ref. 65) is mainly focused on the role of IDH1 in GBM. However, the authors also studied the upregulation of IDH1 in lung cancer and in hematological malignancies (see Supplemental Figure 5).

Line 574, Journal name Plos One is missing.

The reference has been corrected.

Round 2

Reviewer 1 Report

This work is now up to the standards set by the journal.